# Recent Progress on Moisture Absorption Aging of Plant Fiber Reinforced Polymer Composites

**DOI:** 10.3390/polym15204121

**Published:** 2023-10-17

**Authors:** Quan Wang, Tuo Chen, Xiaodong Wang, Yue Zheng, Jiayu Zheng, Gaojie Song, Shuyi Liu

**Affiliations:** College of Mechanical Engineering, Tianjin University of Technology and Education, Tianjin 300222, China; chentuo202209@163.com (T.C.); wangxdong1997@163.com (X.W.); jacqueline112588@163.com (Y.Z.); zhengjy1029@163.com (J.Z.); sgj15866882935@163.com (G.S.); 18822215612@163.com (S.L.)

**Keywords:** plant fiber, moisture absorption performance, water absorption kinetics, mechanical properties, biodegradability

## Abstract

Plant fiber reinforced polymer matrix composites have attracted much attention in many industries due to their abundant resources, low cost, biodegradability, and lightweight properties. Compared with synthetic fibers, various plant fibers are easy to obtain and have different characteristics, making them a substitute for synthetic fiber composite materials. However, the aging phenomenon of composite materials has been a key issue that hinders development. In natural environments, moisture absorption performance leads to serious degradation of the mechanical properties of composite materials, which delays the use of composite materials in humid environments. Therefore, the effects of moisture absorption performance of plant fiber composite materials on their mechanical properties have been summarized in this article, as well as various treatment methods to reduce the water absorption of composite materials.

## 1. Introduction

In the past few decades, synthetic fiber (glass fiber, carbon fiber, etc.) reinforced polymer matrix composites have been used in various industries. However, most synthetic fibers have non-renewable and non-biodegradable properties, which affect the natural environment and promote the gradual replacement of elements of synthetic fiber composites by plant fiber composites [1,2]. Compared with synthetic fiber composites, plant fiber composites also have certain limitations, such as significant poor interfacial adhesion between plant fibers and polymers, mainly due to the interaction between the polarity and hydrophilicity of plant fibers and the non-polarity and hydrophobicity of polymers. Therefore, it there is an urgent need to improve the adhesion between plant fibers and polymers and reduce the hygroscopicity of plant fiber composite materials. The most efficient measure to improve the moisture absorption of plant fibers is to modify their surface through physical/chemical methods [3,4,5,6,7]. Chemical methods have mainly used chemical reagents to treat hemicellulose, lignin, and amorphous areas of plant fibers, which are the main sources of moisture absorption. In addition to using chemical treatment methods, polymer coatings have been added to the surface of plant fibers [8,9,10]. Physical methods such as fiber blending are also important methods for reducing the moisture absorption properties of plant fibers [11,12,13,14,15,16,17].

At present, researchers have not only conducted extensive research on the hygroscopic properties of plant fiber reinforced polymer matrix composites, but also reported the impact of hygroscopic properties on the mechanical properties of composite materials [18,19]. The impact of modified plant fiber composites on water absorption and moisture absorption and the mechanical properties of composite materials have been summarized in this paper.

## 2. Water Absorption Mechanism

The mechanical properties and dimensional stability of plant fiber composite materials were found to be adversely affected due to prolonged exposure to humid environments [10]. Therefore, the water absorption mechanism is crucial for improving the durability of composite materials. The water absorption property accelerates the expansion of plant fibers, leading to the degradation of the fiber matrix interface. Ultimately, this will lead to a decrease in the mechanical properties and change in size of the composite material [10,13,20,21,22,23,24,25]. The sequence of composite material failure caused by plant fibers due to water absorption is shown in Figure 1. Azwa et al. [21] introduced the mechanism of the influence of moisture absorption properties on the fiber matrix interface, as shown in Figure 2. The permeation pressure increases after water-soluble substances leach from the fiber surface, which will lead to debonding between the fiber and the matrix. The moisture absorption properties of plant fiber composite materials are usually studied according to the standard of ASTMD 570 [26], and the water absorption rate is as follows [13,27].
(1)$$W\%=\frac{W_{2}-W_{1}}{W_{1}}\times 100\%$$
where, *W*_1_ and *W*_2_ are the initial mass before soaking and the final mass after soaking, respectively.

In recent years, researchers have developed different models to characterize the water absorption behavior of composite materials, but the overall approach is to model by considering diffusion mechanisms. The diffusion of water molecules into composite materials can be explained according to the Fickian diffusion law. At room temperature, the water absorption process of composite materials is very similar to the Fickian diffusion process [12,21,22,28,29,30,31]. At higher temperatures, non Fickian diffusion processes [32] are observed as mass transfer between the two media by changing the position of molecules under the influence of thermal energy, concentration gradients, electricity, magnetism, and stress.

The diffusion types of plant fiber composite materials are shown in Table 1 [12,21,32,33], and the intercept and slope of the water absorption curve are plotted to distinguish the diffusion types as follows:(2)$$\frac{M_{a}}{M_{b}}=kT^{n}$$
(3)$$\log\left(\frac{M_{a}}{M_{b}}\right)=\log k+n\log T$$
where, *T* is time, k is the constant of the interaction between the composite material spline and water, and n is the constant of the diffusion mode.

## 3. Water Absorption Behavior of Plant Fibers and Substrates

The high water absorption behavior between plant fibers and the matrix is the main drawback of plant fibers and the main reason for the decrease in mechanical properties of composite materials. The hydrophilicity of plant fibers is due to the presence of hydrogen bonds between cell wall molecules in the fibers. When the fibers come into contact with water molecules, the hydrogen bonds break and—OH groups form new hydrogen bonds with water molecules. Therefore, it can be concluded that hydrophilic—OH groups are the main source of fiber water absorption, and reducing the water absorption behavior of composite materials requires the elimination of hydrophilic groups. Due to the differences in plant fibers, their internal structures are diverse, and their inherent hydrophilic chemical components make it easy for them to absorb water during the soaking process, with varying water saturation levels. Table 2 shows the water saturation rates of some plant fibers.

Although the water absorption rate of polymer matrix in plant fiber composite materials is far lower than that of fibers, the diffusion of water molecules in the matrix still conforms to the Fickian diffusion law, and the hygroscopic properties also affect the mechanical properties of the polymer matrix. The mechanism of the effect of hygroscopic properties on the polymer matrix is as follows: (1) Water molecules enter the matrix, which forces the distance between polymer molecular chains to increase, causing the matrix to swell and the polymer to undergo plasticization; (2) As the environmental temperature increases, the relaxation of polymer molecular chains intensifies, which weakens the intermolecular forces and accelerates the formation of molecular gaps within the polymer. The ability of water molecules to diffuse towards the polymer matrix is enhanced, and the water absorption rate and saturation amount are both increased; (3) The diffusion of water molecules towards the polymer matrix results in the formation of new cracks within the matrix due to the osmotic pressure generated. The continuous diffusion and increase of cracks further increase the water absorption capacity of the polymer matrix; (4) High temperature water molecules undergo hydrolysis reactions with hydrophilic groups such as amide groups, ether bonds, and amine groups in the polymer matrix, resulting in chain segments and cross-linking. The soaking time of different polymer matrices in water plays a crucial role in achieving water absorption saturation, as shown in Figure 3. Compared with different thermosetting polymer resins, epoxy resin has the lowest water absorption rate and the best water absorption stability with increasing soaking time. Compared with different thermoplastic polymer resins, low-density polyethylene and polypropylene have the lowest water absorption rate, while PA6 and PA66 have overall higher water absorption rates, When the immersion time of high-density polyethylene in water increases from 24 h to 48 h, its water absorption varies greatly [34,35]. It has been found that the tensile and flexural strength of both single plant fiber composite materials and multiple plant fiber composite materials continuously decrease with the prolongation of soaking time, resulting in a continuous decrease in mechanical properties. Maslinda, A.B. et al. found that the mechanical properties of composite materials show a significant downward trend with the continuous increase of moisture absorption [36].

## 4. Effect of Hygroscopic Properties on Mechanical Properties of Composite Materials

The hygroscopic properties of plant fiber composite materials have a negative impact on their mechanical properties, and their industrial application performance is not good. Therefore, it is necessary to study the mechanical properties of composite materials after water absorption. The influence of moisture absorption properties on mechanical properties of sisal fiber (SF) composite materials are shown in Figure 4 [18,37]. Due to the moisture absorption properties of the composite material, the fibers expand and cause debonding between the fibers and polymers, which results in a decrease in the mechanical properties of the composite material. The tensile strength and modulus decrease by 11.69–22.77% and 12.75–30.51%, respectively. The bending strength and modulus decrease by 11.72–27.34% and 5.47–35.9%, respectively. Shahzad, A. et al. [38] studied the effect of hygroscopic properties on the mechanical properties of hemp fiber (HF) composite materials and found that hygroscopic properties led to a significant decrease in the tensile and flexural strength of hemp fiber composite materials. Moreover, when the composite materials were soaked in water for about 2000 h, the tensile and flexural strength lost about 30% and 65%, respectively. Chow, C. et al. [39] found that the tensile strength and Young’s modulus of sisal fiber composites continuously decrease with increasing immersion time. On the contrary, the impact strength initially increases with increasing immersion time. After reaching the maximum impact strength, the impact strength decreases with increasing immersion time, mainly due to the plasticization of the composite interface and the expansion of sisal fibers, This has led to an increasing trend in the impact strength of composite materials. Sodium bicarbonate (NaHCO_3_) has been used to treat the fibers and a mixed NaHCO_3_ and polylactic acid coating to improve the water resistance and mechanical properties of sisal fiber (SH) composite materials [19]. Research has found that the water absorption rate of the mixed treated sisal fiber composite material was 30% lower than that of the untreated one, and the hygroscopic properties of the mixed treated sisal fiber composite material had the smallest impact on the bending strength and tensile strength, with a decrease of 3.59% and 33.18% in tensile strength and bending strength, respectively. From the microstructure, it can be found that the interface adhesion strength of the unmodified sisal fiber composite material was poor, and the polymer matrix cracks. The fiber surface was smooth and the cross-section was flat, and there was a debonding phenomenon within the polymer matrix. However, the fiber surface of NaHCO_3_ modified composite materials and mixed modified composite materials was relatively rough and there were polymer matrix particles, indicating that the adhesive strength between the modified fibers and the polymer matrix interface increases and there was no fiber debonding phenomenon. Gupta, M.K. et al. [9] studied the aging behavior of short cut jute fiber/polylactic acid (SJF/PLA) composite materials in a humid and hot environment. The composite materials were prepared by a lamination hot pressing method, and a comparative study was conducted on uncoated samples and samples coated with PP film in steam at 70 °C. It was found that PP film coating can effectively delay the water absorption aging process of the composite materials. Hu, R.H. et al. [40] used 90 °C hot water to soak different mass fractions of sisal fiber/polypropylene composites and found that, as the soaking time increased, the mechanical properties of the composite continued to decrease, and the higher the sisal fiber mass fraction, the faster the rate of decline in the mechanical properties of the composite. Moudood, A. et al. [41] studied the mechanical properties of linen fiber/epoxy resin (FF/Epoxy) composite materials in different environments. Compared with composite materials placed in normal environments, they found that the tensile strength and flexural strength of the composite materials exposed to humid environments decreased by 9% and 64%, respectively.

As shown in Table 3 and Figure 5, the lowest decrease in tensile strength of composite materials after water absorption was observed in flax fiber composite materials and hemp fiber composite materials, with a decrease of 11.27% and 12.7%, respectively. Comparing with Zone II and Zone IV, it was found that the composite materials prepared by mixing the two plant fibers did not effectively improve their tensile strength after water absorption. As shown in to Zones I and VI, it had been proven that composite materials prepared by mixing synthetic fibers and plant fibers can effectively improve their tensile strength after water absorption, with an increase of 12.58% in tensile strength. Compared with plant fibers, glass fibers had lower hygroscopicity and higher mechanical properties. In the mixed preparation, the decrease in the mass fraction of plant fibers and the increase in the mass fraction of synthetic fibers promotes the continuous decrease in hygroscopicity and improvement of mechanical properties of composite materials. In Zone III, it was found that NaHCO_3_ improved the surface modification of plant fibers, resulting in a 7% increase in the tensile strength of the composite material after water absorption. Additionally, the combination of NaHCO_3_ and PLA coating improved the tensile strength of the composite material after water absorption by 9.14%.

As shown in Table 3 and Figure 6, the flax fiber composite material and banana fiber (BF) composite material showed the lowest decrease in bending strength after moisture absorption, with a decrease of 7.26% and 9.26%, respectively. Comparing Zone I and Zone V, as well as Zone II and Zone IV, it was found that the composite material prepared by mixing the two plant fibers did not effectively improve the bending strength of the composite material after water absorption, However, the hybrid preparation of synthetic fibers and plant fibers can effectively improve the flexural strength of composite materials after water absorption, resulting in a 12.55% increase in flexural strength after water absorption. In summary, the mechanical properties of hemp fiber composite materials after water absorption had the smallest decrease, and were influenced by various factors. Composite materials prepared by mixing synthetic fibers and plant fibers had the most significant improvement effect on their mechanical properties after water absorption.

## 5. Methods for Reducing the Water Absorption of Composite Materials

In order to address the adverse effects of natural fiber moisture absorption on the mechanical properties and dimensional stability of composite materials, researchers have modified fibers through various chemical and physical methods. The various methods to reduce the water absorption of composite materials are introduced in this section, such as fiber surface chemical treatment, the use of compatibilizers, fiber mixing, nanofillers, and polymer coatings.

### 5.1. Chemical Methods

#### 5.1.1. Alkali Treatment

Alkali treatment is one of the most common chemical modification methods, that works mainly by breaking the plant fiber OH group and reacting with water molecules to remove it from the plant fiber structure. It can also eliminate hemicellulose, lignin, and wax in plant fibers. This method can increase the surface roughness of plant fibers and improve the adhesion strength between fibers and polymers to minimize the water absorption of composite materials. Gunti Rajesh et al. [47] modified short sisal fibers with 10% NaOH and H_2_O_2_ (hydrogen peroxide), and prepared modified and unmodified short sisal fiber reinforced polylactic acid (PLA). The composite material showed a sharp increase in the first 24 h, followed by a stable trend, and the water absorption rate of the composite material gradually increased with the increase of plant fiber content. Compared with the unmodified short sisal fiber composite material, the water absorption rate of the modified composite material was much lower.

#### 5.1.2. Benzoylation

Benzoyl chloride in benzoylation mainly reduces the hydrophilicity of plant fibers and enhances the adhesion strength between plant fibers and polymers, where hydroxyl groups remove water absorbing substances such as wax and lignin through the fiber surface. Sreekumar, P.A. et al. [48] used various chemical methods to treat fibers, such as 100 °C heat treatment, potassium permanganate treatment, benzoylation treatment, silane treatment, and sodium hydroxide treatment, and prepared composite materials using resin transfer molding (RTM). The adsorption coefficients of sisal fiber reinforced polyester matrix composite material is shown in Table 4. After modification, the water absorption rate of the fiber reinforced polyester matrix composite material was greatly reduced. As the impregnation temperature increases, the adsorption coefficient of the composite material also continuously increases, leading to a decrease in the bonding strength of the fiber matrix interface and the formation of microcracks and voids in the composite material. This greatly increases the water absorption saturation of the composite material. Therefore, the change in temperature greatly affects the water absorption efficiency of fibers and the water storage capacity of composite materials. The order of water absorption of modified composite materials was as follows: unmodified > 100 °C heat treatment > silane treatment > permanganate treatment > sodium hydroxide treatment > benzoylation treatment.

#### 5.1.3. Silane Coupling Agent

Silane coupling agent modification is one of the most effective methods to improve the interfacial adhesion of plant fiber composite materials. Cui, Y.H. et al. [49] studied the modification of wood fibers with silane coupling agents and prepared wood fiber/recycled polypropylene (rPP) composite materials. Compared with untreated wood fiber composite materials, they found that the water absorption of the treated composite materials decreased, mainly due to the hydrolysis, condensation, and bonding stages of the fibers during the silane coupling agent treatment process. During the hydrolysis stage, silane forms silane alcohol in the presence of fiber moisture, one end of the silane alcohol reacts with the matrix functional group, and the other end reacts with the cellulose hydroxyl group. This stage prevented the fibers from expanding into the hydrocarbon chains in the matrix and resulted in molecular continuity at the interface of the composite material and, ultimately, improved the adhesion between the fibers and the matrix.

#### 5.1.4. Maleic Anhydride

Maleic anhydride treatment of plant fibers mainly reacts with—OH groups present in the amorphous region of the fiber cell wall, and reduces the hydrophilicity of plant fibers by forming a long polymer chain coating. Maleic anhydride grafted polypropylene (MAPP) has been used to treat loofah fiber (LF)/polypropylene composite materials [50]. The study showed that the addition of maleic anhydride grafted polypropylene improved the interface between fibers and polymers, not only limiting the diffusion of water molecules in the composite material, but also reacting with—OH groups present in the amorphous region of the fiber cell wall. This eliminated—OH from the fiber, reduced the hydrophilicity of plant fibers, and reduced the water absorption of composite materials.

#### 5.1.5. Acetylation

Acetylation treatment is used to change the structure of fibers, and is also known as the esterification method. In acetylation treatment, a reaction occurs between the acetyl group (-CH_3_C0) and hydroxyl group (-OH) of the fiber to eliminate water. Bledzki, A.K. et al. [51] used acetylation treatment on flax fiber and found that the interfacial adhesion of flax fiber/polypropylene composite material was enhanced. The acetylation treatment of flax fiber changed the structure of cellulose and caused a reaction between the acetyl group (CH300) and hydroxyl group (-OH) of the fiber to eliminate water, thereby reducing the hydrophilicity of the fiber and enhancing the moisture resistance of the composite material. Various chemical reagents have been used to modify the surface of plant fibers, mainly to maximize the treatment of hemicellulose, lignin, and cellulose parts of plant fibers. These substances reduced the water absorption of composite materials after it was treated with chemical reagents.

#### 5.1.6. Sodium Bicarbonate

Roy, J.K. et al. [52] used sodium bicarbonate (NaHCO_3_) to modify short jute fibers and prepared short jute fiber reinforced polypropylene composites. The study found that the water absorption of the composite material significantly decreased after NaHCO_3_ modification of short jute fibers. Fiore, V. et al. [53] prepared epoxy resin based composites by treating sisal fibers with 10 wt% sodium bicarbonate solution at room temperature. Their study found that the interface adhesion strength and mechanical properties of the composite materials were best when treated with sisal fibers for 120 h. With a longer treatment time, the surface roughness of sisal fibers significantly increased, a large amount of impurities in the fibers were eliminated, and the diameter of all treated fibers significantly decreased. Due to the release of lignin and hemicellulose from the fibers, their internal structure was caused to rearrange, form high-density dense structures, and fibers with lower moisture absorption.

### 5.2. Fiber Hybrids

Fiber hybrid reinforced polymer composites are composite materials with two or more different fiber reinforced polymers. The main purpose of introducing another fiber type into a single composite material is to maintain the advantages of the two hybrid fibers and reduce the disadvantages of a single fiber. As shown in the Figure 7, there are three types of fiber hybrid structures. Figure 7a shows the compression of single fibers into blocks and cross placement, Figure 7b shows the cross weaving of two fibers, and Figure 7c shows the direct mixing of the two fibers. Plant fiber hybrid reinforced polymer composites are mainly divided into two types, plant fiber hybrid and plant fiber and synthetic fiber hybrid [54].

#### 5.2.1. Plant Fiber Hybrid

As shown in Table 5, Maslinda, A.B. et al. prepared hybrid composite materials with different plant fibers, and found that the tensile strength and bending strength of the composite materials decreased significantly after moisture absorption. The moisture absorption rate of different plant fiber hybrid composite materials mainly depends on the moisture absorption rate of different plant fibers [36].

Gunturu, B. et al. [55] studied the effects of banana fiber (BF), coconut shell fiber (coir), and hybrid fibers on the water absorption of composite material, as shown in Figure 8. They found that coconut shell fiber reinforced polypropylene composite material had the highest water absorption, and the water absorption of the composite material increased significantly in the first 24 h. Vallejos, M.E. et al. [56] tested the water absorption of composite materials by immersing the fibers in distilled water and placing them in an environment with a relative humidity of 50%. They found that the higher the mass fraction of hemp fiber, the stronger the water absorption ability. The composite material soaked in distilled water reached water absorption saturation after 25 days, and the composite material soaked in 50% relative humidity reached water absorption saturation after 40 days. Hernández Jiménez, J.A. et al. [57] studied the water absorption of wood fiber reinforced polypropylene composites and found that the composite had the highest water absorption when the wood fiber particles were fifty mesh. Rotich Gideon et al. [58] studied the moisture absorption properties of jute/palm leaf fiber (palm) hybrid composites and demonstrated that the higher the mass fraction of jute fiber, the stronger the water absorption ability of the composite. From the perspective of water absorption, different fiber blends can improve the water resistance of the composite material, and it can be seen that palm leaf fiber has a lower water absorption rate, and the composite material can improve moisture resistance.

#### 5.2.2. Synthetic Fiber Hybrid

Mixing natural fibers with synthetic fibers (such as glass fibers and carbon fibers) are an effective method to solve the water absorption of composite materials. Mixing carbon fibers and glass fibers into plant fibers can improve the water resistance and mechanical properties of composite materials [59,60,61,62,63]. Therefore, an increase of hydrophobic fibers can reduce the moisture content and absorption rate of the composite material [43,64,65,66,67,68]. Hassan et al. [69] produced a mixed composite material of jute fiber and glass fiber through manual stacking method. Table 6 and Table 7 show the effects of different stacking sequences and fiber content ratios on the mechanical properties and water absorption performance of composite material. Research has found that with the intervention of glass fibers, fiber hybridization improves the tensile and bending strength of composite materials and, as the proportion of glass fibers continues to increase, the water absorption performance of the composite material continues to decrease. This is mainly because the water absorption and mechanical properties of synthetic fibers are much better than those of plant fibers. Increasing the proportion of synthetic fibers in fiber mixing leads to a continuous decrease in the water absorption performance and improvement in the mechanical properties of composite materials. Thwe, M.M. et al. [15] used MAPP to hybrid modify glass fiber/bamboo fiber to enhance PP composite material. The composite material was immersed in water for 1200 h and, compared with the unmixed composite material, the water absorption rate of the hybrid fiber composite material decreased by 4%. Panthapulakkal, S. et al. [25] found that the mixing of hemp fiber and glass fiber significantly reduced the water absorption of the composite material.

### 5.3. Polymer Coating

Some researchers have attempted to coat polymers on the surface of fibers to improve the water resistance of composite materials, enhance interfacial bonding ability, and reduce the water absorption of composite materials. Rodriguez et al. [8] used a combination of alkali treatment and polyhydroxybutyrate (PHB) coating treatment on jute fibers and found that the treated jute fiber composite material had higher mechanical properties and water resistance. The alkali treatment and coating treatment overcame the limitations of jute fiber modified composite materials, and this new method maximized the water resistance of the composite material. As shown in Figure 9 and Figure 10, Gupta, M.K. et al. [10] used alkali treatment, polylactic acid coating treatment, and mixed treatment to enhance the water absorption of polyester composite materials on the surface of jute fibers. They compared the water absorption of treated and untreated jute fiber composite materials and found that the water absorption of jute fiber mixed treatment composite materials was lower, at 4.02%.

Wu, Y. et al. [70] used the Walli process to in situ adhere polyethylene (PE) film to the surface of hemp fiber composite materials, as shown in Figure 11. This technology used PE film instead of traditional vacuum bags to further produce polyethylene plant fiber composite materials (PE–NFRC). Compared with untreated composite materials, the water absorption of PE–NFRC was significantly reduced by 88.5%, 84.8%, and 68.67% at 2, 24, and 120 h, respectively. The results showed that the surface coating of PE film significantly improved the water resistance of the composite material.

### 5.4. Nanofillers

Nanofillers such as nanoclay and nano silicon carbide can be added to natural fiber composite materials to improve the water absorption and mechanical properties of the composite material in humid environments. A nanoclay layer forms an impermeable medium to prevent the flow of water molecules, allowing for longer diffusion time of water molecules, and reduces the water absorption of the composite material. Azam, F.A. et al. [71] found that the water absorption of kenaf fiber/polypropylene composites without the addition of nanomaterials was lower than that of composites with the addition of multi walled carbon nanotubes (MWCNTs). Majeed, K. et al. [72] studied the effect of maleic anhydride grafted polypropylene (PP–g–MAH) and nano montmorillonite (MMT) on the water absorption of rice husk fiber (RH)/polypropylene composite materials. They found that the addition of PP-g–MAH and MMT can reduce the water absorption of the composite material. On the one hand, PP–g–MAH improved the compatibility of the composite material, reduced micropores in the composite material, and reduced the water absorption of the composite material. On the other hand, MMT is a waterproof nanomaterial, and the addition of PP–g–MAH and MMT simultaneously exhibited the lowest water absorption.

Liu, Y. et al. prepared a SiO_2_/jute fiber/polypropylene composite through melt gel technology and compression molding, as shown in Figure 12 [73]. The influence of SiO_2_ on the moisture absorption and interface failure of the composite were studied. Through molecular dynamics simulation, it was found that SiO_2_ effectively protected the composite from the influence of water diffusion by strong molecular chain interlocking on the interface of jute fiber/polypropylene composite, The water resistance and dimensional stability of the composite material were improved, and the water absorption rate and thickness expansion rate of the composite material were reduced by 3.6% and 3.0%, respectively.

As shown in Table 8 and Figure 13, it was found that the water resistance of composite materials prepared by mixing plant fibers and synthetic fibers was higher than that of composite materials prepared by mixing two types of plant fibers. Moreover, polymer coating on the fiber surface greatly improved the water resistance of the composite material, which was much higher than that of chemically modified plant fiber composite materials. Compared with PLA coated composite materials, the water resistance of PE film composite materials prepared by the Vary process was higher. Compared with chemically modified composite materials, the water resistance of composite materials prepared by mixing chemical modification and nanomaterials was higher. As shown in Figure 13, the three best methods for water resistance were: mixing plant fibers with synthetic fibers to prepare composite materials, preparing PE membrane composite materials using the Walli process, and preparing composite materials using chemical modification and nanofillers. Generally, the water resistance of composite materials prepared by mixing the two processes was higher than that prepared by a single process.

## 6. Conclusions

(1)The water absorption of composite materials is affected by various parameters, such as the selection of polymers and plant fibers, the mass fraction of plant fibers, fiber modification methods, water absorption time, and environmental humidity. However, the main parameters are the selection of plant fibers, fiber mass fraction, and water absorption time. It was concluded that plant fiber and synthetic fiber hybrid, polymer coating, and tile process coating have significantly improved the water resistance of composite materials, based on the current research.(2)The modification of the plant fiber polymer interface has always been a hot topic in the research on composite materials. Currently, there are many modification methods emerging, but the key technology for modifying plant fiber by polymer coating and Walli process coating still lags behind that in foreign countries. In addition, using these methods alone to treat plant fiber and its composite material’s water resistance cannot achieve the optimal state, Therefore, mixing polymer coating and Vary process coating with other physical/chemical methods for modification is an important research direction for improving the water resistance of plant fiber composite materials in the future.(3)Whether plant fiber composite materials can replace synthetic fiber composite materials in certain fields mainly depends on the changes in mechanical properties of plant fiber composite materials after water absorption. The authors believe that modifying plant fibers through various physical/chemical methods and different preparation processes can improve the mechanical properties of composite materials, but it cannot improve the moisture absorption properties of composite materials any further, ultimately leading to poor mechanical properties after water absorption. Therefore, it is necessary to develop a universal and low-cost technology to jointly improve the mechanical properties and water resistance of composite materials.(4)The researchers have developed water molecule diffusion models to study the water absorption behavior of composite materials and explain the water resistance performance of different composite materials. Therefore, developing different models to study the water resistance performance of composite materials is also an important direction for research in the future. By developing different water absorption models to express the water absorption behavior of plant fiber composite materials and continuously promote the development of new molding and modification processes, efforts will be made to achieve the development of high intelligence and low production costs, thereby promoting the widespread application of plant fiber composite materials.

## Figures and Tables

**Figure 1 polymers-15-04121-f001:**
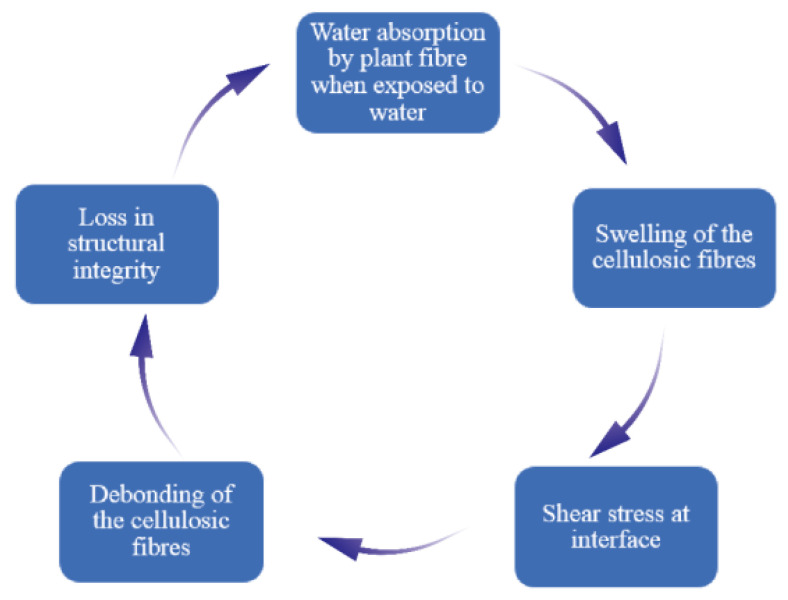
Composite material failure caused by water absorption.

**Figure 2 polymers-15-04121-f002:**
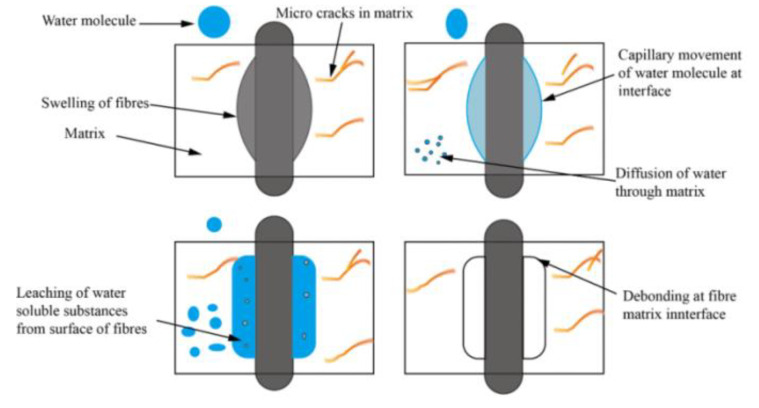
Fiber debonding phenomenon caused by water absorption [2].

**Figure 3 polymers-15-04121-f003:**
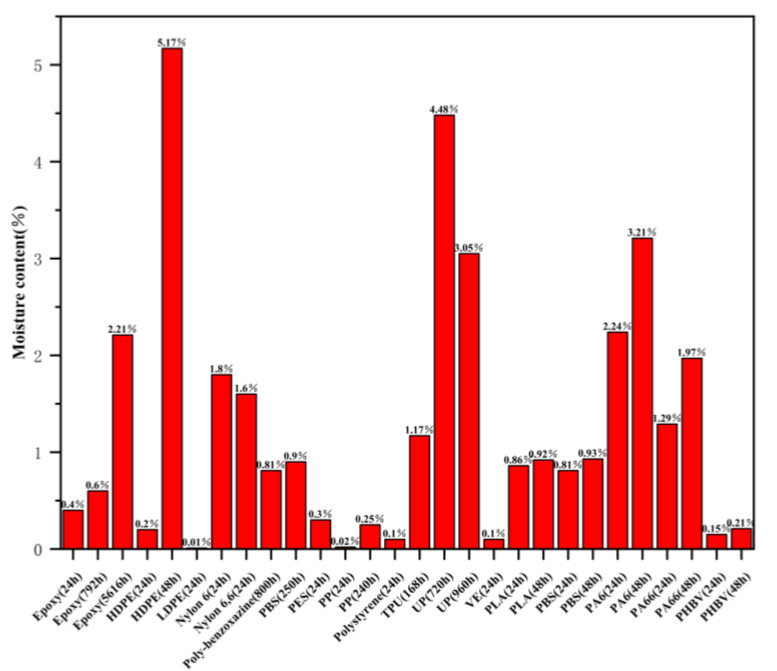
Water absorption performance of different polymer materials.

**Figure 4 polymers-15-04121-f004:**
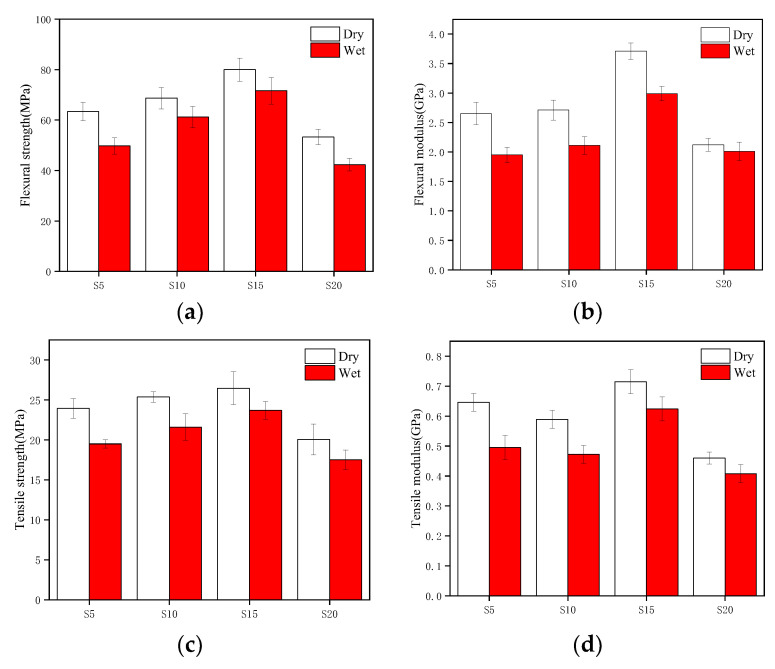
Comparison of mechanical properties of composite materials before and after water absorption: (**a**) flexural strength (**b**) flexural modulus (**c**) tensile strength (**d**) tensile modulus [18].

**Figure 5 polymers-15-04121-f005:**
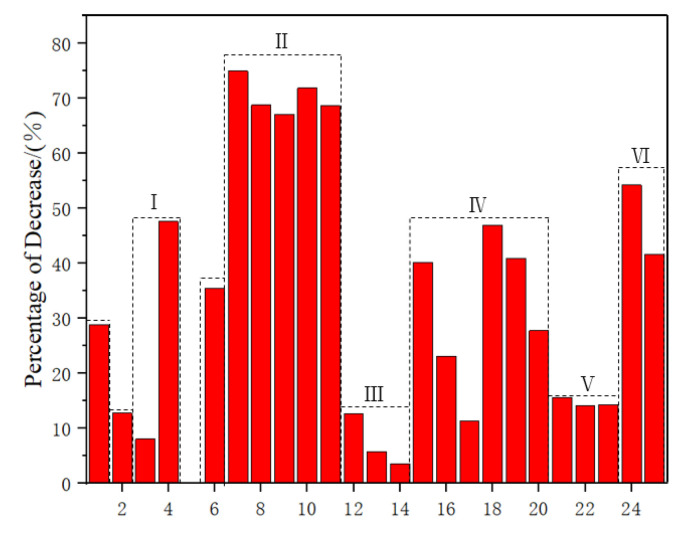
Percentage decrease in tensile strength of composite materials after water absorption. I: Pennisetum/Glass–Epoxy; II: Kenaf/Jute/Hemp–Epoxy; III: Sisal/Epoxy; IV: Jute/Hemp/Flax–Epoxy; V: Sisal/Epoxy; VI: Jute/Glass –Polyester.

**Figure 6 polymers-15-04121-f006:**
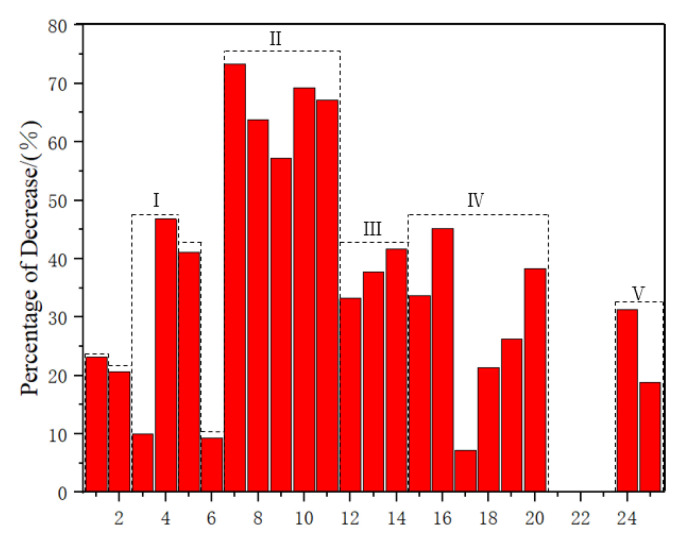
Percentage decrease in bending strength of composite materials after water absorption. I: Pennisetum/Glass–Epoxy; II: Kenaf/Jute/Hemp–Epoxy; III: Sisal/Epoxy; IV: Jute/Hemp/Flax–Epoxy; V: Jute/Glass–Polyester.

**Figure 7 polymers-15-04121-f007:**
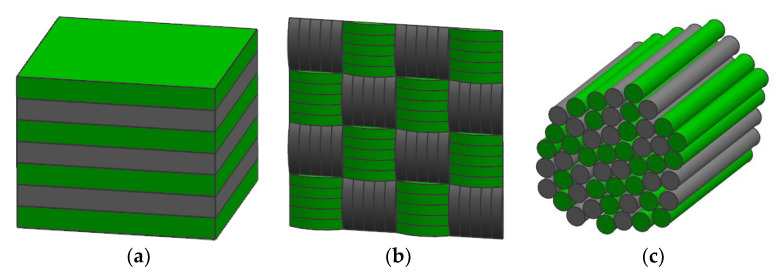
The main hybrid composites configurations: (**a**) interlayer (**b**) intralayer and (**c**) intrayarn.

**Figure 8 polymers-15-04121-f008:**
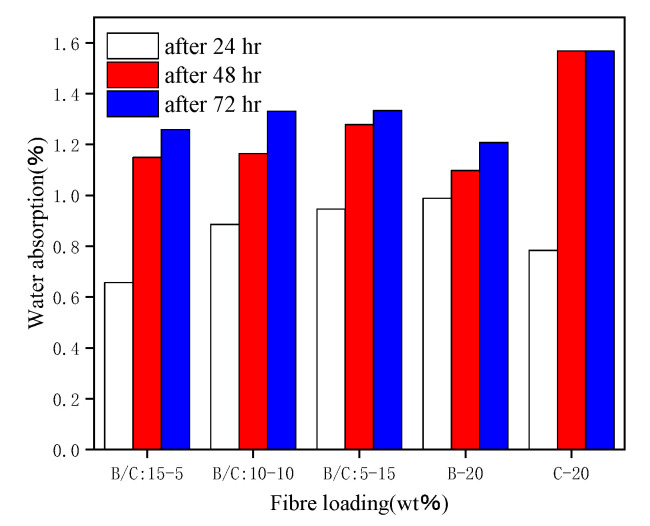
Comparison of water absorption at different times.

**Figure 9 polymers-15-04121-f009:**
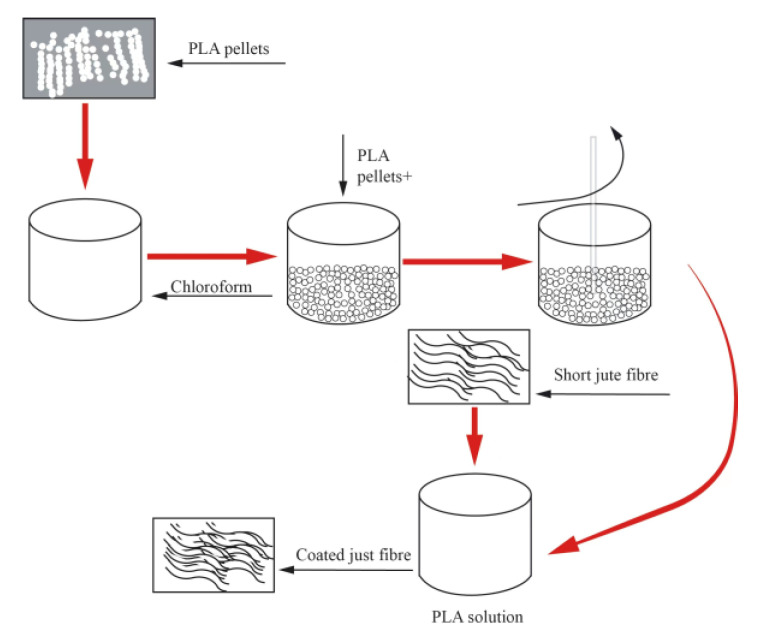
Surface coating treatment of jute fiber with polylactic acid.

**Figure 10 polymers-15-04121-f010:**
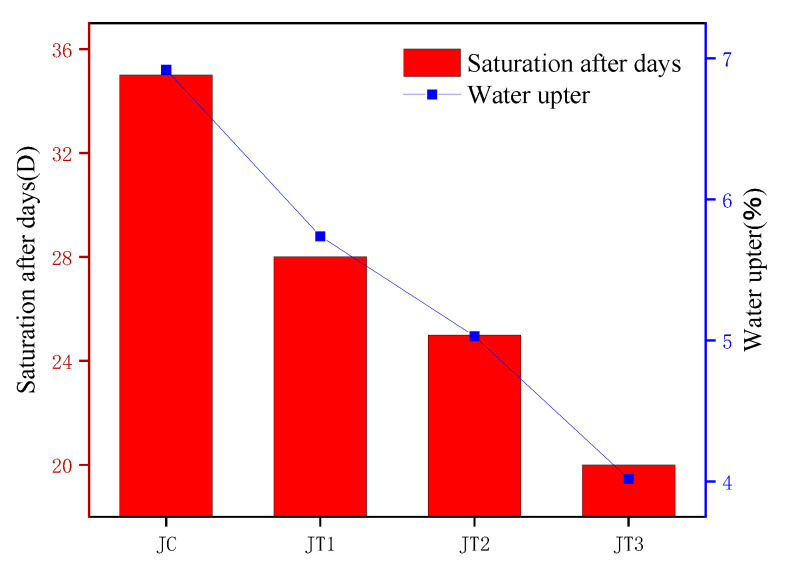
Comparison of the effects of different treatment methods on the water absorption performance of composite materials.

**Figure 11 polymers-15-04121-f011:**
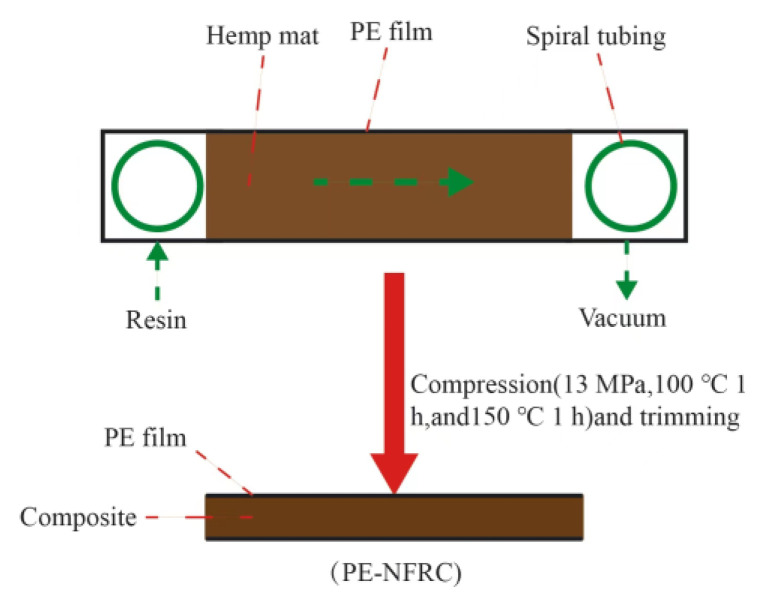
Schematic diagram of the Walli process.

**Figure 12 polymers-15-04121-f012:**
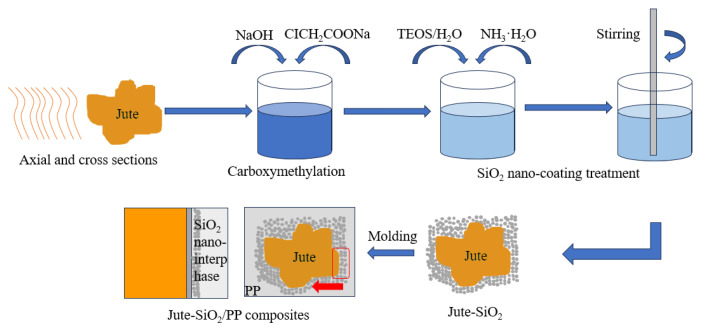
Schematic of the synthesis procedures for jute–SiO_2_/PP composites via the sol-gel technique and compression molding.

**Figure 13 polymers-15-04121-f013:**
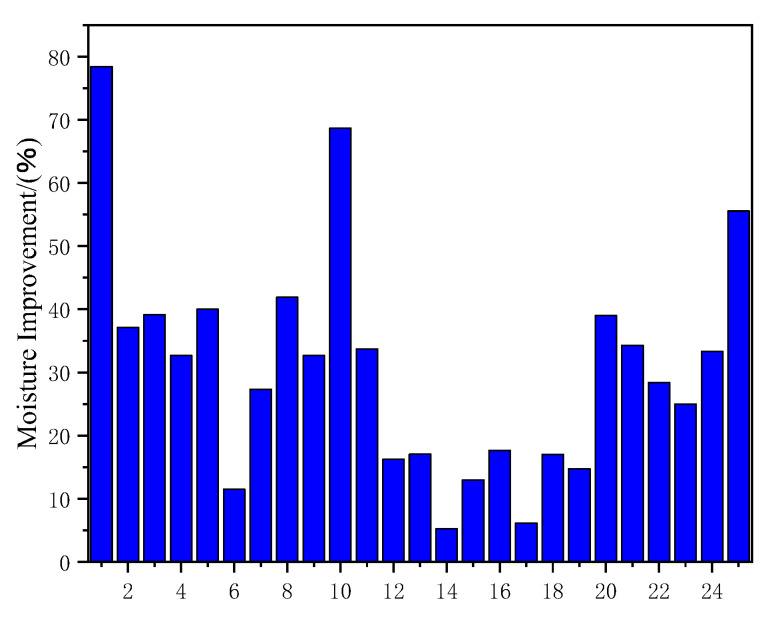
Improving water absorption performance of composite materials with different treatment methods.

**Table 1 polymers-15-04121-t001:** Water diffusion mechanism of pure and modified sisal composites.

Diffusion Type		Diffusion Index (*n*)	Time	Mechanization
Case1	Less Fickian	*n* < 0.5	$t^{-1/2}$	Rate of diffusion of water molecules is much less than that of polymer segment mobility
	Fickian Diffusion	*n* = 0.5	$t^{1/2}$	
Case2	Case2 Diffusion	*n* = 1.0	(time independent)	Diffusion process is much more active than relaxation processes
	Super Case2 Diffusion	*n* > 1.0	$t^{n-1}$	
Case3	Non-Fickian/ Anomalous Diffusion	0.5 < *n* < 1.0	$t^{n-1}$	Mobility of water molecules is comparable to that of polymer segment mobility; It is an intermediary performance between Case1 and Case2 diffusion

**Table 2 polymers-15-04121-t002:** Saturated water absorption of different natural fibers at 65% RH (relative humidity).

Fiber	Water Absorption Saturation Rate/%
Flax	7
Hemp	7
Abaca	7
Agave	8
Sisal	7

**Table 3 polymers-15-04121-t003:** Comparison of tensile strength and bending strength of composite materials before and after water absorption.

Number	Composite	Note	Tensile Strength (MPa)		Decline Rate	Bending Strength (MPa)		Decline Rate	References
			Before Absorbing Water	After Absorbing Water		Before Absorbing Water	After Absorbing Water		
1	Waste Paper/Polyester		70.2	50	28.77%	96.76	74.41	23.10%	[42]
2	Sisal/Epoxy		20.06	17.51	12.72%	53.26	42.3	20.58%	[18]
3	Pennisetum/		21–131	11–120.52	8–47.62%	47–125	25–112.5	10–46.81%	[43]
4	Glass–Epoxy								
5	Cotton fabric/		-	-	-	15.8	9.3	41.14%	[44]
	Geopolymer								
6	Stem banana Polypropylene		38.55	24.91	35.38%	55.59	50.44	9.26%	[45]
7	Kenaf/Jute/	K/epoxy	80	20	75%	77.6	20.7	73.32%	[36]
	Hemp–Epoxy								
8		J/epoxy	77	24	68.83%	68.5	24.8	63.80%	
9		H/epoxy	76	25	67.11%	68.2	29.2	57.18%	
10		K/J/epoxy	89	25	71.91%	95	29.2	69.26%	
11		K/H/epoxy	83	26	68.67%	90.8	29.8	67.18%	
12	Sisal/Epoxy		26.49	23.13	12.70%	60.5	35.3	33.18%	[19]
13		NaHCO3	27.1	25.55	5.70%	67.6	42.14	37.67%	
14		NaHCO3/PLA coating	35.5	34.23	3.56%	71.5	47.77	41.65%	
15	Jute/Hemp/	J/epoxy	43.32	25.9	40.21%	59.47	39.44	33.68%	[46]
	Flax–Epoxy								
16		H/epoxy	36.68	28.2	23.12%	85.59	46.92	45.18%	
17		F/epoxy	46.21	41	11.27%	81.1	75.21	7.26%	
18		J/H/epoxy	42.19	22.4	46.90%	86.6	68.11	21.35%	
19		H/F/epoxy	44.17	26.1	40.91%	44.6	32.89	26.26%	
20		J/H/F/epoxy	58.59	42.3	27.80%	66.6	41.1	38.29%	
21	Sisal/Epoxy		45	38	15.56%	-	-	-	[29]
22		nanoclay	57	49	14.04%	-	-	-	
23		microclay	49	42	14.29%	-	-	-	
24	Jute/Glass	J/polyester	135.53	62.02	54.24%	115.61	79.38	31.34%	[12]
	–Polyester								
25		J/G/polyester	261.22	152.4	41.66%	366.38	297.54	18.79%	

**Table 4 polymers-15-04121-t004:** Adsorption coefficient of sisal fiber reinforced polyester matrix composites at different temperatures.

Composite (mol%)	Temperature (°C)	Adsorption Coefficient (Q∞)	Composite (mol%)	Temperature (°C)	Adsorption Coefficient (Q∞)
R40	30	0.6409	RB40	30	0.2329
	60	0.8390		60	0.2429
	90	0.8080		90	0.3263
RN40	30	0.3124	RP40	30	0.3433
	60	0.3283		60	0.3856
	90	0.4068		90	0.4236
RH40	30	0.4931	RS40	30	0.3526
	60	0.5762		60	0.3829
	90	0.6352		90	0.4009

**Table 5 polymers-15-04121-t005:** Comparison of water absorption and mechanical properties of composite materials before and after water absorption.

Properties		Specimens				
		KK	JJ	HH	KJ	KH
Tensile strength (MPa)	Dry	80 ± 4	77 ± 3	76 ± 5	89 ± 3	83 ± 1
	Saturated	20 ± 4	24 ± 1	25 ± 3	25 ± 2	26 ± 1
Flexural strength (MPa)	Dry	77.6 ± 2.9	68.5 ± 5.3	68.2 ± 4.9	95 ± 3.4	90.8 ± 6.3
	Saturated	20.7 ± 3.6	24.8 ± 3.8	29.9 ± 2.3	29.2 ± 5.1	29.8 ± 2.7
Moisture absorption rate (%)		14.1	13.9	12.1	7.5	5.1

**Table 6 polymers-15-04121-t006:** Water absorption results of the composites (J: jute fiber; G: glass fiber).

Sample Name	Fiber Orientation	Water Absorption (%)				
		24	96	120	168	192
C1	JJJJJ	7.43	7.5	7.7	7.7	7.7
C2	JJGJJ	6.26	6.6	6.7	6.8	6.8
C3	JGJGJ	5.58	5.7	5.8	5.8	5.8
C4	GJGJG	4.41	4.9	5	5	5
C5	GGJGG	3.39	3.7	3.8	3.8	3.8
C6	GGGGG	2.1	2.3	2.4	2.4	2.4

**Table 7 polymers-15-04121-t007:** Results of the tensile, flexural, and microhardness tests of the composite.

Sample Name	Fiber Orientation	Tensile Strength (MPa)	Flexural Strength (MPa)	Hardness HV
C1	JJJJJ	38.6875	54.71	13
C2	JJGJJ	59.375	80.4	20.8
C3	JGJGJ	64.025	85.5	27.7
C4	GJGJG	104.625	134.65	32.2
C5	GGJGG	92.1	125.95	34.5
C6	GGGGG	106.8	176.8	39.9

**Table 8 polymers-15-04121-t008:** Effects of different treatment methods on improving water absorption performance of composite materials.

	Composite	Fiber Treatment	Water Absorption Rate		Rising Rate	References
			Untreated	After Treatment		
1	Pennisetum purpureum /Glass-Epoxy	-	7.00%	1.51%	78.43%	[43]
2	Hemp/Glass-Polypropylene	-	8.73%	5.49%	37.11%	[74]
3	Coir/Glass-Polyester	-	8.53%	5.19%	39.16%	[75]
4	Wood/Hemp-Polypropylene	-	26.00%	17.50%	32.69%	[64]
5	Jute/Palm-Recycled Polypropylene	-	1.25%	0.75%	40.00%	[59]
6	Banana/Sisal-Epoxy	-	21.25%	18.81%	11.48%	[29]
7	Jute-Polyester	PLA coating	6.92%	5.03%	27.31%	[10]
8	Jute-Polyester	alkali treatment/PLA coating	6.92%	4.02%	41.91%	[10]
9	Sisal-Polyester	PLA coating	5.66%	3.80%	32.68%	[76]
10	Hemp-Polyethylene	PE flim	33.20%	10.40%	68.67%	[9]
11	Jute-Polylactic acid	PP coating	17.50%	11.60%	33.71%	[40]
12	Sisal-Polyester	NaOH treatment	5.66%	4.74%	16.25%	[9]
13	Jute-Polyester	alkali treatment	6.92%	5.74%	17.05%	[77]
14	Kenaf-Polypropylene	NaOH/KMno4	21.51%	16.27%	5.24%	[40]
15	Rice husk-Polypropylene	PP-g-MA	11.39%	9.91%	12.99%	[50]
16		PP-g-MA/SEBS-g-MA	-	9.38%	17.65%	
17		Silane	-	10.69%	6.15%	
18		Silane/PP-g-MA	-	9.45%	17.03%	
19		Silane/PP-g-MA/SEBS-g-MA	-	9.71%	14.75%	
20	Luffa fibre /Polypropylene	(3-Aminopropyl) triethoxysilane (AS)	2.80%	1.71%	39.00%	[73]
21		3-(Trimethoxysilyl)-1-propanethiol (MS)	-	1.84%	34.30%	
22		Maleic anhydride grafted PP (MAPP)	-	2.00%	28.40%	
23	Rice husk /Polypropylene	PP-g-MAH	0.36%	0.27%	25.00%	[72]
24		MMT	-	0.24%	33.33%	
25		PP-g-MAH/MMT	-	0.16%	55.55%	

## Data Availability

All data, models, and code generated or used during the study appear in the submitted article.

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
