# Peer review of "Recent Progress on Moisture Absorption Aging of Plant Fiber Reinforced Polymer Composites"

_polymers, 2023, doi:10.3390/polym15204121_

Round 1
Reviewer 1 Report
This paper summarizes a large amount of work reported in the literature. It is helpful to the reader to get this overview. Different fibers and their treatments are compared with respect to water absorption and sometimes to strength and stiffness.
Figure 2 shows a degradation mechanism leading to fiber matrix debonding. It is a bit unclear whether this mechanism is applicable to all materials reviewed or only some. Please clarify.
Page 3 (Equations 2 to 10) explaining diffusion could probably deleted. This information is not used later anymore and it is probably known to most readers. When describing 3-D diffusion some other works should be cited, that have, for example, shown that diffusion along fibers can be much faster than diffusion transverse to the fibers.
Section 3:
This section seems talks about water absorption but seems to mix water absorption rate and saturation level. Usually only the saturation level is of interest, because that influences the mechanical properties in the long run. How quickly that happens (absorption rate) is far less important, at least if saturation is reached within the lifetime of the structure. Please make very clear what effect is compared.
General (all section):
The paper sometimes compares absorbed water after 100 and 200 hours. Unfortunately, such data are often reported in the literature, but these data have very limited value if the reader does not know whether saturation was reached or not. A critical review should address this issue.
It is stated that fibers absorb much more water than the matrix. This is true in many cases, but generally it depends on the matrix used in the composite.
Figure 4. A reference is missing, or was this work done by the authors?
Section 4 and 5
These sections show many comparisons. It would be helpful if the reader is guided into what would be the expected result. For example, a hybrid composite would probably absorb water based on a rule of mixture of the two types of fibers used. Do the reported results confirm this or do they deviate?
What would be expected for stiffness and strength? Note that for hybrid stiffness usually follows the rule of mixtures, but strength does not.
General:
When adding synthetic fibers and special chemicals to modify fibers and interfaces, the environmental benefits and recyclability may get reduced. This should be commented on.
Conclusions
This paper seems to be 90% a review and a bit of own work. It should be more clear what is what. Maybe make this paper a 100% review paper?
Mostly good
Author Response
This paper summarizes a large amount of work reported in the literature. It is helpful to the reader to get this overview. Different fibers and their treatments are compared with respect to water absorption and sometimes to strength and stiffness.
Figure 2 shows a degradation mechanism leading to fiber matrix debonding. It is a bit unclear whether this mechanism is applicable to all materials reviewed or only some. Please clarify.
Answer: I think it is suitable for most plant fibers
Page 3 (Equations 2 to 10) explaining diffusion could probably deleted. This information is not used later anymore and it is probably known to most readers. When describing 3-D diffusion some other works should be cited, that have, for example, shown that diffusion along fibers can be much faster than diffusion transverse to the fibers.
Answer: Equations 2 to 10 have been deleted.
Section 3:
This section seems talks about water absorption but seems to mix water absorption rate and saturation level. Usually only the saturation level is of interest, because that influences the mechanical properties in the long run. How quickly that happens (absorption rate) is far less important, at least if saturation is reached within the lifetime of the structure. Please make very clear what effect is compared.
Answer: The section 3 mainly represents the differences in water absorption behavior between different plant fibers and matrices.
It is stated that fibers absorb much more water than the matrix. This is true in many cases, but generally it depends on the matrix used in the composite.
Figure 4. A reference is missing, or was this work done by the authors?
Answer: The reference has been added.
Section 4 and 5
These sections show many comparisons. It would be helpful if the reader is guided into what would be the expected result. For example, a hybrid composite would probably absorb water based on a rule of mixture of the two types of fibers used. Do the reported results confirm this or do they deviate?
What would be expected for stiffness and strength? Note that for hybrid stiffness usually follows the rule of mixtures, but strength does not.
Answer: The reported results confirm this.
When adding synthetic fibers and special chemicals to modify fibers and interfaces, the environmental benefits and recyclability may get reduced. This should be commented on.
Answer: The most synthetic fibers have non-renewable and non-biodegradable properties, which affect the natural environment and promote the gradual replacement of synthetic fiber composites by plant fiber composites in some aspects.
Thank you for your review

Reviewer 2 Report
The authors presented a review on polymer composites, specifically on moisture absorption and the effect on mechanical properties. The manuscript needs revision before publication, considering that there are manuscripts in the literature.
Specific comments:
Authors could add a topic on methodology to inform the review criteria. The literature review can be classified according to its purpose, scope, function, and analysis developed type. The authors could make clear in the methodology each one of the previous points. What purpose? What is the scope? What is the function? What is the intent (is it informative, etc)? In addition, which scientific bases were used for the bibliographic review (Web of Science, Google Scholar, scopus, etc). What are the keyword criteria?
Manuscript comments:
> Page 1. Line 40-45. “At present, researchers have not only conducted extensive research on the hygro.......”. Authors should add references in this paragraph;
> Page 5. Line 117-141. “low-density polyethylene and polypropylene have ............”. Authors should add references in this paragraph;
> Figure 3. It is not clear where the presented results were taken from. For a review article, citations must be reported;
> Authors should add a new topic addressing the effect of plasma treatment on fibers and how it affects moisture absorption;
> 5.1. Chemical method. The authors present treatments applied to natural fibers. However, the review is focused on a single article. For a review article, several articles in the literature on the subject are expected. Authors could add a Table under each chemical treatment topic. With that, cite several references and the main findings;
> Work is being carried out with the use of natural oils to make polymeric composites compatible with natural fibers. However, the authors did not report the effect of oil incorporation on moisture absorption;
> The authors should improve the review on hybrid composites with natural fibers and nanofillers;
> Authors must add a topic about future perspectives, including presenting limitations and advantages;
Author Response
The authors presented a review on polymer composites, specifically on moisture absorption and the effect on mechanical properties. The manuscript needs revision before publication, considering that there are manuscripts in the literature.
Manuscript comments:
> Page 1. Line 40-45. “At present, researchers have not only conducted extensive research on the hygro.......”. Authors should add references in this paragraph;
Answer: The reference has been added.
> Page 5. Line 117-141. “low-density polyethylene and polypropylene have ............”. Authors should add references in this paragraph;
Answer: The reference has been added.
> Figure 3. It is not clear where the presented results were taken from. For a review article, citations must be reported;
Answer: According to the summary of reference data, the figure was made by myself.
> Authors should add a new topic addressing the effect of plasma treatment on fibers and how it affects moisture absorption;
> 5.1. Chemical method. The authors present treatments applied to natural fibers. However, the review is focused on a single article. For a review article, several articles in the literature on the subject are expected. Authors could add a Table under each chemical treatment topic. With that, cite several references and the main findings;
> Work is being carried out with the use of natural oils to make polymeric composites compatible with natural fibers. However, the authors did not report the effect of oil incorporation on moisture absorption;
> The authors should improve the review on hybrid composites with natural fibers and nanofillers;
> Authors must add a topic about future perspectives, including presenting limitations and advantages;
Answer: The authors have made modifications, as detailed in the red section of the paper.
Thank you for your review

Reviewer 3 Report
The manuscript titled "Recent Progress on Moisture Absorption Aging of Plant Fiber Reinforced Polymer Composites" provides valuable insights into the challenges posed by moisture absorption in plant fiber-reinforced polymer composites. However, there are a few areas where it could be improved: The manuscript could benefit from a more explicit statement of its objectives and research questions. What specific aspects of moisture absorption and aging in plant fiber composites are being addressed? While the manuscript mentions summarizing the effects of moisture absorption on mechanical properties and treatment methods, it would be beneficial to provide a more comprehensive review of recent research findings and advancements in this area. Consider refining the structure of the manuscript to clearly delineate sections for the effects of moisture absorption and treatment methods. This will help readers navigate and understand the content more effectively. Ensure that relevant and recent sources are cited to support the information presented. Providing a robust list of references will strengthen the manuscript's credibility. Conclude the manuscript with a concise summary of key findings and their implications for future research or applications in the field of plant fiber-reinforced polymer composites.
Overall, the English in the manuscript is acceptable, but some refinements in sentence structure, word choice, and consistency would enhance its clarity and readability. It's advisable to have the manuscript proofread by a native English speaker or a professional editor to address these minor language issues effectively.
Author Response
The manuscript titled "Recent Progress on Moisture Absorption Aging of Plant Fiber Reinforced Polymer Composites" provides valuable insights into the challenges posed by moisture absorption in plant fiber-reinforced polymer composites. However, there are a few areas where it could be improved: The manuscript could benefit from a more explicit statement of its objectives and research questions. What specific aspects of moisture absorption and aging in plant fiber composites are being addressed? While the manuscript mentions summarizing the effects of moisture absorption on mechanical properties and treatment methods, it would be beneficial to provide a more comprehensive review of recent research findings and advancements in this area. Consider refining the structure of the manuscript to clearly delineate sections for the effects of moisture absorption and treatment methods. This will help readers navigate and understand the content more effectively. Ensure that relevant and recent sources are cited to support the information presented. Providing a robust list of references will strengthen the manuscript's credibility. Conclude the manuscript with a concise summary of key findings and their implications for future research or applications in the field of plant fiber-reinforced polymer composites.
Overall, the English in the manuscript is acceptable, but some refinements in sentence structure, word choice, and consistency would enhance its clarity and readability. It's advisable to have the manuscript proofread by a native English speaker or a professional editor to address these minor language issues effectively.
Answer: Thank you very much for your review. The manuscript has been revised.

Round 2
Reviewer 1 Report
Many previous comments were answered, but a few important once were not addressed sufficiently.
Section 3:
This section seems talks about water absorption but seems to mix water absorption rate and saturation level. Usually only the saturation level is of interest, because that influences the mechanical properties in the long run. How quickly that happens (absorption rate) is far less important, at least if saturation is reached within the lifetime of the structure. Please make very clear what effect is compared.
The paper has not been updated to address this issue.
General (all section):
The paper sometimes compares absorbed water after 100 and 200 hours. Unfortunately, such data are often reported in the literature, but these data have very limited value if the reader does not know whether saturation was reached or not. A critical review should address this issue.
This issue should be explained in the paper and the reader should be made aware of the pitfalls of comparing short time absorption values.
Section 4 and 5
These sections show many comparisons. It would be helpful if the reader is guided into what would be the expected result. For example, a hybrid composite would probably absorb water based on a rule of mixture of the two types of fibers used. Do the reported results confirm this or do they deviate?
What would be expected for stiffness and strength? Note that for hybrid stiffness usually follows the rule of mixtures, but strength does not.
The answer given stated that the results confirm this. Please show this and give the message to the reader in the paper.
Author Response
Section 3:
This section seems talks about water absorption but seems to mix water absorption rate and saturation level. Usually only the saturation level is of interest, because that influences the mechanical properties in the long run. How quickly that happens (absorption rate) is far less important, at least if saturation is reached within the lifetime of the structure. Please make very clear what effect is compared.
Answer: In most papers, it is not possible to determine whether the material has reached water absorption saturation. In an ideal state, composite material water absorption saturation refers to the fact that the material's mass remains unchanged for a period of time during the water absorption experiment, which means it has reached water absorption saturation. However, in practical experiments, no researchers have truly studied the absolute value of water absorption saturation for a composite material. Most of the research mainly focuses on the water absorption rate of composite materials over a period of time, Both water absorption rate and water saturation have an impact on the mechanical properties of composite materials, so this section adds the influence of changes in water absorption time on the mechanical properties of composite materials.
The introduction of relevant parts has been added to the paper.
General (all section):
The paper sometimes compares absorbed water after 100 and 200 hours. Unfortunately, such data are often reported in the literature, but these data have very limited value if the reader does not know whether saturation was reached or not. A critical review should address this issue.
Answer: The water absorption saturation of a material is an absolute value, and it is not known in most of the paper data whether the material has reached water absorption saturation. The chart mainly expresses the comparison of water absorption rates of different materials at 24H and 48h, and as for more comparison of materials at different times, it mainly serves as an introduction.
The introduction of relevant parts has been added to the paper.
Section 4 and 5
These sections show many comparisons. It would be helpful if the reader is guided into what would be the expected result. For example, a hybrid composite would probably absorb water based on a rule of mixture of the two types of fibers used. Do the reported results confirm this or do they deviate?
Answer: In this section, two common mixing rules for fibers have been added. Fiber mixing is mainly divided into two types of plant fiber mixing and plant fiber and synthetic fiber mixing. Firstly, the mixing of two types of plant fibers is undeniable. Therefore, whether the water absorption of composite materials can be improved mainly depends on whether the water absorption of the second type of plant fiber is lower than that of the first type of plant fiber, Less than can improve the overall water absorption of the composite material, while greater than can increase the overall water absorption of the composite material; Secondly, there is a mixture of plant fibers and synthetic fibers. It is evident that the water absorption of plant fibers is much greater than that of synthetic fibers, and the water absorption of synthetic fibers is very small and negligible. Therefore, adding synthetic fibers and mixing them can greatly improve the water absorption and mechanical properties of composite materials. However, plant fibers and synthetic fiber composite materials must be water absorbing, The size of water absorption mainly depends on the proportion of synthetic fibers.
The introduction of relevant parts has been added to the paper.
What would be expected for stiffness and strength? Note that for hybrid stiffness usually follows the rule of mixtures, but strength does not.
Answer: The accurate predicted values of the strength and stiffness of composite materials, as well as the predicted values of mixed composite materials, have not been reported in relevant literature.
Thank you very much for your review.
Reviewer 2 Report
The authors improved the quality of the manuscript. In addition, recommendations have been accommodated in the revised version. In view of this, the manuscript can be published.
Author Response
Thank you very much for your review.
Round 3
Reviewer 1 Report
the last comments were reasonably answered.
This reviewer still thinks that properties should be compared for saturated materials and the absorption rates are of secondary importance. But if the reviewed papers lack this information, not much can be done about it. This is the reason for a somewhat low "star rating".